# Clinical Veterinary Boron Neutron Capture Therapy (BNCT) Studies in Dogs with Head and Neck Cancer: Bridging the Gap between Translational and Clinical Studies

**DOI:** 10.3390/biology9100327

**Published:** 2020-10-07

**Authors:** Amanda E. Schwint, Andrea Monti Hughes, Marcela A. Garabalino, Gustavo A. Santa Cruz, Sara J. González, Juan Longhino, Lucas Provenzano, Paulina Oña, Monica Rao, María de los Ángeles Cantarelli, Andrea Leiras, María Silvina Olivera, Verónica A. Trivillin, Paula Alessandrini, Fabricio Brollo, Esteban Boggio, Hernan Costa, Romina Ventimiglia, Sergio Binia, Emiliano C. C. Pozzi, Susana I. Nievas, Iara S. Santa Cruz

**Affiliations:** 1National Atomic Energy Commission (CNEA), Avenida del Libertador 8250, C1429 BNP, Buenos Aires, Argentina; andre.mh@gmail.com (A.M.H.); marcegarabalino@gmail.com (M.A.G.); termografia.bnct@gmail.com (G.A.S.C.); srgonzal@gmail.com (S.J.G.); longhino@cab.cnea.gov.ar (J.L.); lucasprovenzano@hotmail.com (L.P.); marisil.olivera@gmail.com (M.S.O.); verotrivillin@gmail.com (V.A.T.); brollonono@gmail.com (F.B.); efboggio@gmail.com (E.B.); eccpozzi@gmail.com (E.C.C.P.); susanaisabelnievas@gmail.com (S.I.N.); iarasofiasantacruz@gmail.com (I.S.S.C.); 2National Research Council (CONICET), Godoy Cruz 2290, C1425FQB, Buenos Aires, Argentina; 3Fundación INTECNUS: Instituto de Tecnologías Nucleares para la Salud, Ruta Provincial 82, San Carlos de Bariloche, R8402AGP, Provincia Rio Negro, Argentina; paulina.ona@intecnus.org.ar (P.O.); costahernan06@gmail.com (H.C.); romina.ventimiglia@intecnus.org.ar (R.V.); sergiobinia@gmail.com (S.B.); 4Hospital Veterinario, Gobernador M. Ugarte 2152, Olivos, B1636BWT, Provincia Buenos Aires, Argentina; poro70@hotmail.com; 5Independent Veterinarian, Huilqui 12356, San Carlos de Bariloche, 8400, Provincia Rio Negro, Argentina; m.andrealeiras@gmail.com; 6Independent Veterinarian, Lonquimay 3817, San Carlos de Bariloche, 8400, Provincia Rio Negro, Argentina; paualessandrini@gmail.com

**Keywords:** Boron Neutron Capture Therapy, BNCT, clinical veterinary BNCT studies, head and neck cancer, veterinary medicine

## Abstract

**Simple Summary:**

Boron Neutron Capture Therapy (BNCT) is a treatment for cancer based on the selective accumulation in tumor of boron compounds, followed by external irradiation with neutrons. The interaction between boron-10 and a neutron gives rise to very energetic particles that travel only a very short distance (approximately the diameter of a cell) and are lethal for the cell. In this way, BNCT damages tumor tissue selectively while preserving normal tissue. BNCT has proved effective to treat certain tumors in clinical trials worldwide, with room for improvement. Our group has worked on animal models to improve the efficacy of BNCT, in particular for head and neck cancer. Herein we performed clinical veterinary BNCT studies in five terminal dog patients with head and neck cancer with no other therapeutic option. In all cases we observed partial tumor response, clinical benefit, and extension of estimated survival time at recruitment with excellent quality of life. Toxicity associated to the treatment was mild/moderate and reversible. These studies contribute towards preparation for clinical BNCT trials for head and neck cancer in Argentina and suggest a potential role for BNCT in veterinary medicine.

**Abstract:**

Translational Boron Neutron Capture Therapy (BNCT) studies performed by our group and clinical BNCT studies worldwide have shown the therapeutic efficacy of BNCT for head and neck cancer. The present BNCT studies in veterinary patients with head and neck cancer were performed to optimize the therapeutic efficacy of BNCT, contribute towards exploring the role of BNCT in veterinary medicine, put in place technical aspects for an upcoming clinical trial of BNCT for head and neck cancer at the RA-6 Nuclear Reactor, and assess the feasibility of employing the existing B2 beam to treat large, deep-seated tumors. Five dogs with head and neck cancer with no other therapeutic option were treated with two applications of BNCT mediated by boronophenyl-alanine (BPA) separated by 3–5 weeks. Two to three portals per BNCT application were used to achieve a potentially therapeutic dose over the tumor without exceeding normal tissue tolerance. Clinical and Computed Tomography results evidenced partial tumor control in all cases, with slight-moderate mucositis, excellent life quality, and prolongation in the survival time estimated at recruitment. These exploratory studies show the potential value of BNCT in veterinary medicine and contribute towards initiating a clinical BNCT trial for head and neck cancer at the RA-6 clinical facility.

## 1. Introduction

Boron neutron capture therapy (BNCT) is classically described as a biologically tumor cell-targeted therapy for cancer. It is a binary treatment modality that involves the selective accumulation of boron carriers in tumors, followed by irradiation with a thermal or epithermal neutron beam. The high linear energy transfer α particles and recoiling ^7^Li nuclei emitted during the capture of a thermal neutron by a ^10^B nucleus have a range of 5–9 μm in tissue and are known to have high relative biological effectiveness. In this way, BNCT would potentially target neoplastic tissue effectively and selectively, while preserving healthy tissue [1]. Since BNCT involves biological rather than geometrical tumor targeting, it is also ideally suited to treat undetectable micrometastases, infiltrating malignant cells and foci of malignant transformation in field-cancerized tissue, with minimum damage to healthy tissues in the treatment volume [2,3].

Our main interest is focused on BNCT for head and neck tumors. Head and neck malignancies are often radio-/chemo-resistant and undergo extensive growth, requiring mutilating surgeries [4]. Translational studies by our group in the hamster cheek pouch oral cancer model [5,6,7,8,9,10], exploring what was then a new target for BNCT, and clinical studies worldwide [11,12,13] have shown the therapeutic efficacy of BNCT for head and neck cancer while preserving oro-facial structures and functions. BNCT studies by our group in appropriate animal tumor models seek to contribute to the knowledge of BNCT radiobiology, optimize the therapeutic efficacy of BNCT for existing and novel targets for BNCT, and design safe and effective BNCT protocols. A large part of our efforts have focused on developing strategies to optimize BNCT for head and neck cancer, employing boron compounds already approved for use in clinical trials. This approach reduces the gap between translational research and clinical application.

Studies in in vivo animal tumor models are pivotal to the advancement of BNCT. However, admittedly, small animal models fail to recapitulate the full complexity of human tumors. Companion animals, mainly dogs and cats, offer translational models that mimic more adequately the intricate nature of spontaneous neoplasia in humans [14]. Additionally, the body size difference between humans and rodents makes these models often incapable of accurately predicting therapeutic efficacy and toxicity in human patients. The dog’s head size makes the complex dosimetric variables relevant to human radiation therapy [15].

In general, dogs and cats develop similar cancers at analogous sites to those in humans. Like humans, dogs and cats are outbred animals, and share the same environment. The fact that animal cancers progress rapidly is an advantage in terms of the study periods that are necessary to obtain results. The biological behavior and response to treatment of many spontaneous malignancies in dogs and cats are similar to those in humans. Within this context, spontaneously occurring tumors in pet dogs are considered appropriate and valid model tumor systems to test therapeutic strategies [15,16,17,18]. Studies in dog patients (albeit with constraints) would mimic a clinical scenario and provide an adequate link between our translational research and a future clinical trial.

Furthermore, the role of BNCT in veterinary medicine in and of itself warrants evaluation and analysis. Cancer is a major cause of death in pets [19]. Dogs develop tumors twice as frequently as humans (although cats only half as frequently). One study reported that 45% of dogs that lived to 10 years or more died of cancer. Regardless of age, 23% of the dogs assessed in this study died of cancer [20]. Because there are few established treatments in veterinary medicine, this is an area where BNCT studies are particularly welcome, both as a contribution to veterinary medicine itself and as an input for clinical trials.

Within this context, BNCT studies in dog and cat patients with tumors are a valuable contribution to clinical BNCT trials and to veterinary medicine, and constitute an intermediate stage between translational and clinical studies, improving the safety and efficacy of clinical protocols. 

We performed exploratory BNCT studies in dog patients with head and neck cancer with no other therapeutic option at the RA-6 Nuclear Reactor BNCT clinical facility in San Carlos de Bariloche (Argentina). The aims of these studies were to further contribute to the knowledge of BNCT and optimize its therapeutic efficacy for head and neck cancer, contribute towards exploring the role of BNCT in veterinary medicine, put in place the technical aspects for a future clinical trial of BNCT for head and neck cancer at the RA-6 Nuclear Reactor, and assess the feasibility to treat large, deep-seated tumors employing the existing B2 beam, tuned to treat superficial tumors. Although the B2 beam was originally designed for the treatment of superficial lesions, its applicability could conceivably be extended to deeper-seated targets using suitable planning strategies. While the BNCT project in Argentina has experience in a clinical BNCT trial for peripheral melanoma [21], technical aspects and requirements vary according to the pathology and location under evaluation and must be addressed accordingly.

The malignancies that are considered appropriate for therapeutic study in dogs and cats are osteosarcoma, mammary neoplasia, oral melanoma, oral squamous cell carcinoma (SCC), lung cancer, soft tissue sarcomas, and malignant non-Hodgkin’s lymphoma [16,19,22]. In the particular case of BNCT, a therapy designed for the local treatment of solid tumors, the potential targets under consideration would not include systemic diseases. Because our particular interest is centered on BNCT for head and neck cancer, we enrolled five dog patients with head and neck cancer with no other therapeutic option in a clinical veterinary exploratory BNCT study at RA-6.

## 2. Materials and Methods

Consent was obtained from the dog’s owner in each case. In addition, treatment protocols were approved by the National Atomic Energy Commission Animal Care and Use Committee (CICUAL-CNEA 01/2015; 01/2018_V2). Five dog patients with head and neck cancer with no other therapeutic option were enrolled in the study according to the following inclusion criteria:Confirmed histopathological diagnosis of head and neck cancer;Not amenable or unresponsive to alternative standard therapeutic options;Life expectancy with acceptable quality of life (QoL) of more than 1–2 months to ensure minimum follow-up time and clinical/cardiological status to tolerate prolonged (2 h) anesthesia during irradiation for BNCT;If the dog received previous treatment (surgery/chemotherapy), a minimum of 3 weeks should elapse between the end of this treatment and BNCT;No distant metastasis at initial presentation.

A Computed Tomography (CT) scan was performed prior to treatment to determine the volume, extension, and localization of the tumor and proximity to radiosensitive structures. The CT scan was employed for treatment planning. The NCTPlan treatment planning system used for the cutaneous melanoma treatments in Argentina was applied for dose planning [23]. Tissue composition for transport computations was defined according to the ICRU Report 46. Doses to the gross tumor volume (GTV) and to different normal tissues were computed using the whole blood average boron concentration during irradiations and assuming tumor-to-blood, mucosa-to-blood, and other normal tissue-to-blood boron concentration ratios of 3:1, 2:1, and 1:1, respectively. The radiobiological relative biological effectiveness (RBE) and compound biological effectiveness (CBE) weighting factors were used to compute biologically weighted doses. In BNCT radiobiology, the measured relative biological effectiveness factor for the component of the dose from the ^10^B(n,α)^7^Li reaction has been termed the compound biological effectiveness (CBE) factor [1]. CBE values are specific for the boron compound employed and, similarly to RBE, depend on the tissue and endpoint considered. RBE factors were taken as 3.0 for neutron dose and 1 for gamma dose, and CBE factors were taken as 3.8, 2.5, and 1.3 for boron dose in tumor, normal skin/mucosal membranes, and other normal tissues, respectively [21,24]. In the present study, we assumed a mucosa-to-blood ^10^B concentration ratio of 2.0 and a CBE factor for mucosal membrane of 2.5, in keeping with the Finnish protocol for the treatment of head and neck cancer patients [12,24]. Upper dose limits to healthy structures were respected. These were the average dose to the brain of 7 Gy-Eq [25], the maximum dose to the skin of 22 Gy-Eq [26], and the maximum absorbed dose to mucosa of 6 Gy, as established in the Finnish protocol for the treatment of head and neck cancer patients [12]. Based on our NTCP (normal tissue complication probability) model for mucositis G3 or higher, which depends on the four absorbed doses delivered in BNCT rather than on RBE-weighted doses [27], we determined that 6 Gy of total absorbed dose administered with our B2 clinical beam gives a very low probability of radiotoxic effects (NTCP<0.04). Since the FIR1 clinical beam employed in the Finnish protocol has different relative components compared to the B2 clinical beam, the same physical dose of 6 Gy delivered with FIR1 gives a higher NTCP value of about 0.5. Therefore, we decided to limit the absorbed dose to the mucosa to 6 Gy and to then scale this limit based on our NTCP model on a case by case basis. Table 1 and Table 2 show the tumor doses and normal tissue doses, respectively, for the 5 veterinary patients treated with the B2 clinical beam of the RA-6 reactor.

The five dogs with large, deep-seated head and neck tumors with no other therapeutic option were treated with BNCT mediated by the boron compound boronophenyl-alanine (BPA) at the RA-6 Nuclear Reactor (Bariloche Atomic Center) in San Carlos de Bariloche, Argentina (a city approximately 2 h by plane from Buenos Aires where our laboratories are located). If the dogs were recruited in Buenos Aires, they were transported by plane to San Carlos de Bariloche accompanied by their owner. The animals received two BNCT applications administered at 3- to 5-week intervals, using the B2 beam of the Argentine BNCT facility. This beam is mixed thermal–epithermal and was tuned to treat shallow tumors such as cutaneous melanoma. Within this context, treatment planning employed 2–3 portals (according to the case) per BNCT application to achieve a potentially therapeutic dose over the tumor and optimize dose distribution in the Clinical Target Volume, without exceeding normal tissue tolerance. The beam port comprises a protruding conic delimiter of borated polyethylene and lead with a circular aperture 15 cm in diameter, spaced 15 cm from the cone base. At the exit aperture, a flat and radially well-delimited in-air neutron flux distribution is obtained. The in-air thermal and epithermal neutron fluxes, respectively, estimated by computational analysis at the patient position and averaged over the circular aperture of 15 cm are 2.47 × 10^8^ and 2.16 × 10^8^ n cm^−2^ s^−1^ with a statistical uncertainty of less than 1%. In addition, the specific gamma dose, i.e., Dγ/ϕ th+epi, is 1.74 × 10^−12^ Gy cm^2^ n^−1^.

On the day of irradiation, an L-BPA-fructose dose of 350 mg/kg was administered intravenously over a period of 45 min before neutron irradiation. Blood samples collected periodically during and after the infusion of the boron compound were analyzed by inductive coupled plasma (ICP-OES) to estimate boron concentrations as a function of time. An open 2-compartment model was used to fit the data and predict boron concentration in the blood during the irradiation. Tumor boron concentration, however, was not measured (no tissue samples were taken during treatment). To predict the boron dose to the tumor, tumor-to-blood (T/B) ratios were adopted and used. In our case, a T/B value of 3.0 was assumed as a conservative criterion, slightly less than the value used in the Finnish BNCT clinical protocol (T/B = 3.5) and in keeping with translational studies performed by our group in the hamster cheek pouch oral cancer model [5].

The length of irradiation time per portal and initiation time for the first portal (determined in treatment planning) were adjusted based on blood–boron concentration–time profiles. Blood boron concentration values during the irradiation period ranged from 10 to 12 ppm.

Figure 1 depicts an example of our general treatment setting. As can be observed, the blood boron concentration profile shows a bi-exponential decay characterized by a fast and a slow component with different half-life values. Portal irradiations are always performed when the slow component dominates, thus reducing boron concentration changes while maintaining acceptable absolute values.

BNCT treatments were carried out under general anesthesia, a real challenge considering total treatment times of about 2 h (including the irradiation time corresponding to 2–3 portals and positioning prior to each portal) and the need for remote monitoring of vital signs of the patient irradiated inside a closed bunker. We (M.A. Cantarelli) developed, ad hoc, a 10 m esophageal phonendoscope that allows for remote monitoring. Additionally, a camera inside the bunker focused on the breathing bag used for inhalation anesthesia to monitor spontaneous ventilation. The induction was performed by slow intravenous administration of propofol, 2–4 mg/kg. Maintenance was performed with remote inhalation of isoflurane, 1.5–2% in oxygen 100% at an adjustable flow rate of 2 L/minute. The dogs were repositioned between portals as established during treatment planning, employing a laser alignment system. 

Post BNCT, clinical signs were monitored weekly. Tumor response was assessed based on CT studies performed one to three months after BNCT. Potential local radiotoxicity in terms of mucositis was evaluated semi-quantitatively using a 6-point scale (Grade 0–5) according to an adaptation of oral mucositis scales in humans and experimental models [3,28,29]. Oral mucositis is a common, dose-limiting toxicity of radiation and chemotherapeutic antineoplastic therapies for head and neck cancer. It is characterized by breakdown of the oral mucosa and development of ulcerative lesions that result in pain, decreased quality of life, and an increase in the use of healthcare resources. In severe cases, oral mucositis leads to adverse modifications of antineoplastic treatment [30]. Other symptoms of toxicity such as somnolence were also monitored. 

## 3. Results

Table 3 shows a summary of the patients treated, histopathological diagnosis, complementary treatments, tumor response, toxicity, and clinical evaluation. Available tumor volume values calculated from CT scans pre- and post-treatment are also presented. 

In general, all patients were either unresponsive or not amenable to standard therapy. Within that context, they were recruited for BNCT. At the time of recruitment, all five patients were considered terminal with an estimated life expectancy with a reasonable quality of life of 1–2 months. After BNCT, they all exhibited positive tumor response, clinical benefit with mild or, at most, manageable, reversible acute mucositis. In two cases, we observed early reversible, mild somnolence. No mid-long-term toxicity or clinically relevant complications were observed post BNCT. Macroscopic evaluation and CT scans performed 1–2 months post treatment revealed a reduction in tumor volume to 5%–50% of the pre-treatment tumor volume, evidencing partial responses in all cases. Survival with good QoL ranged from 8.5 to 13.5 months after the first application (except in the case of Mora I, who had an impressive local response but was euthanized due to lung metastases 2.5 months after the second application). BNCT prolonged the expected survival of 1–2 months and markedly improved QoL. Very early—as early as 1–7 days after the second application of BNCT—positive response to treatment contributed to a virtually immediate and marked improvement in the general condition of the patients and in their QoL. Figure 2 illustrates an example of a follow-up CT scan 1 month post BNCT for Lucy, showing a 50% reduction in tumor volume.

Figure 3 shows examples of patients pre and post BNCT as indicated.

In the particular case of Lucy, after an initial positive response (Figure 2), tumor regrowth occurred at 10 months after treatment and the patient was re-treated with full dose BNCT (2 applications 3 weeks apart). An initial positive tumor response was observed 2 weeks after BNCT re-treatment, coupled to mild somnolence, mild mucositis, and eye irritation that required medication. Toxicity induced by re-treatment was milder, or at most, similar to that observed after the first treatment. Recurrence and decline occurred 2.5 months after re-treatment.

The applicability of the B2 beam, originally designed for the treatment of superficial lesions, could conceivably be extended to deeper-seated targets using suitable planning strategies such as combined portals. However, the low minimum tumor doses reached with the B2 beam in large superficial and deep-seated lesions would be the cause of recurrence and supports modifications to obtain a more penetrating beam. Figure 4 shows, as an example, the tumor dose distribution for Senshi employing the B2 beam. The fact that the mean tumor dose is close to the maximum value (Table 1) confirms that a significant portion of the tumor volume received high doses. However, since the tumor is deep-seated and the thermal neutron flux of the B2 mixed beam peaks at about 1 cm depth and then decreases, the most distal parts of the lesion received only moderate doses that would not be enough to induce complete tumor remission.

## 4. Discussion

Studies in canine patients with spontaneous tumors that are unresponsive or not amenable to standard treatment pave the way from translational research to clinical trials. Within this context, we performed exploratory BNCT studies in dog patients with head and neck tumors with no other therapeutic option, building on our translational BNCT studies in the hamster cheek pouch oral cancer model [5,6,7,8,9,10] and preparing for a BNCT clinical trial for head and neck cancer in the near future at the RA-6 clinical facility in Argentina. Complementary interest lay in contributing towards establishing an evidence-based role for BNCT in veterinary medicine based on the present study and previous BNCT studies by our group in cat patients with head and neck cancer [31,32]. In this sense, we also hoped to provide contributory data as a basis for future controlled studies in centers worldwide, capable of comparing the efficacy of BNCT vs. Stereotactic Radiation Therapy and Microbeam Radiation Therapy.

The present BNCT study enrolled terminal dogs with large, deep-seated head and neck tumors that were unresponsive or not amenable to standard therapy, i.e., the patients/tumors that are most difficult to treat. In all cases, we observed a positive response of the tumor, with a reduction in tumor volume to 5%–50% of the pre-treatment tumor volume 1 to 2 months post BNCT. Survival post BNCT (8.5–13.5 months) exceeded the estimated survival time (1–2 months) in all cases except one that was euthanized due to lung metastases. Effective and fast tumor response as early as 1–7 days after the second application of BNCT led to a prompt and impressive improvement in quality of life that persisted for several months. Acute toxicity was mild–moderate and reversible. No mid-term toxicity or clinically relevant complications were observed. In the case of Lucy, full dose BNCT re-treatment 10 months after the first treatment induced partial tumor remission, clinical benefit, and mild toxicity that did not exceed that seen after the first treatment. A similar response was observed in a previously reported case of a feline patient with head and neck cancer after full dose BNCT re-treatment (albeit at low dose), 7.5 months after the first treatment [31]. Our experience, admittedly limited, with BNCT re-treatment at full dose suggests the feasibility of re-treating with BNCT at full dose without exceeding radiotolerance. However, it must be noted that tumor response was less durable after re-treatment, conceivably due to regrowth from potentially underdosed and more resistant tumor cell populations. 

Clinical benefit and tumor response were followed by eventual tumor regrowth rather than by complete tumor remission. This finding could be attributed to the fact that in-depth radiation dose was insufficient in large tumors. A similar observation was made by Kato et al. [11] who performed several (two to three) BNCT applications in recurrent head and neck malignancies due to insufficient in-depth doses in large tumors. Although the B2 beam was designed for the treatment of superficial lesions, the use of combined portals could conceivably extend its use to deeper-seated targets. While the present study showed the feasibility of using the B2 beam with adequate treatment planning strategies to treat large, superficial, and deep-seated tumors, the low minimum tumor doses reached would be at the root of the tumor regrowth observed. Although the number of dogs treated is insufficient to draw conclusions in this sense, irradiation with the B2 beam, designed to treat superficial tumors, might lead to underdosing of certain tumor areas, resulting in incomplete tumor remission and regrowth.

This finding supports the convenience of performing modifications to obtain a more penetrating beam prior to initiating clinical BNCT trials for head and neck cancer at the RA-6 clinical facility. Within this context, different beam shaping assembly models and configurations, including delimiters and shields, have been evaluated, all of them aimed at modifying the neutron beam spectrum of the BNCT facility of the RA-6 reactor to obtain an epithermal beam, with a maximum yield of thermal neutrons at greater depths. The computational models also aim at providing a tunable beam by means of moderating interchangeable materials to lower the neutron energies and maintain the original spectrum, originally optimized to treat shallow lesions. Radiation dose must be delivered to all tumor cell populations within a heterogeneous tumor to a therapeutically optimal level. In the case of BNCT, therapeutic, homogenous dose delivery to the tumor depends on beam spectrum/neutron fluence, treatment planning, and tumor boron targeting. Within this context, potential strategies have been postulated to improve boron-10 targeting to tumor cells. The combined administration of ^10^B compounds with different properties and complementary uptake mechanisms improves boron targeting homogeneity in the tumor and enhances the therapeutic effect of BNCT [33]. In the hamster cheek pouch oral cancer model, BNCT mediated by the combined administration of BPA, transported by the L-type amino acid transport system 1 (LAT1) that is overexpressed in tumor cells [34], and the diffusive, non-tumor selective boron compound decahydrodecaborate (GB-10), proved highly effective without radiation-induced toxicity [6] and warrants evaluation in clinical veterinary BNCT studies. Despite the fact that GB-10 does not target hamster cheek pouch tumors selectively, GB-10-BNCT selectively damages the more radiosensitive aberrant tumor blood vessels, sparing precancerous and normal tissue vessels. In the case of GB-10-BNCT, selective tumor lethality would result from selective tumor aberrant blood vessel damage rather than from selective tumor uptake of the boron compound [6]. The fact that both compounds have already been used as stand-alone boron compounds in humans would expedite the extrapolation to a clinical scenario. Other therapeutic strategies, such as Sequential BNCT (BPA–BNCT followed by GB-10–BNCT 24 or 48 h later) [7], aberrant tumor blood vessel normalization prior to administration of BPA for BPA–BNCT [8], and electroporation combined with GB-10–BNCT [35], have improved the therapeutic efficacy of BNCT without associated toxicity. Assessment of these strategies in a clinical veterinary scenario would be contributory.

This clinical veterinary BNCT study in dogs with head and neck cancer is a follow-on study from our translational studies in the hamster cheek pouch oral cancer model and a contribution to upcoming clinical BNCT studies for head and neck cancer in Argentina. The size of the animal, the spontaneous nature of the tumor, and the effect on surrounding tissues adequately mimic a clinical scenario and could contribute towards optimizing a clinical trial [15]. Dogs have previously been used in BNCT-related studies to evaluate the RBE of an epithermal beam for brain tissue [36], the tolerance of normal canine brain to epithermal neutron irradiation in the presence of BPA [37], and normal tissue tolerance with BNCT mediated by sodium borocaptate (BSH) [38,39]. Kraft et al. [40,41] used dogs with spontaneous brain tumors to evaluate BSH biodistribution and implications for BSH-mediated BNCT. These authors stressed the relevance of studying spontaneous tumors as opposed to experimental tumors. The latter may not predict the behavior of their natural counterpart due to differences in vascular permeability and blood flow, kinetics, and tumor morphology. Mitin et al. [42] compared the efficacy of BNCT and GdNCT (Gadolinium-mediated neutron capture therapy) for canine oral melanoma and osteosarcoma and Takeuchi [38] studied the possible application of BSH–BNCT in spontaneous canine osteosarcoma. All of these studies have proved contributory to the knowledge of BNCT.

The present study focuses on BNCT for head and neck cancer in dog patients, contributing towards preparation for clinical trials in Argentina and providing contributory evidence to establish a role for BNCT in veterinary medicine.

## 5. Conclusions

Exploratory clinical veterinary BNCT studies in five dog patients with head and neck cancer with no other therapeutic option showed, in all cases, partial tumor response, clinical benefit, and extension of estimated survival time at recruitment. Toxicity associated to the treatment was mild/moderate and reversible. These studies contribute towards preparation for clinical BNCT trials for head and neck cancer in Argentina and suggest a potential role for BNCT in veterinary medicine.

## Figures and Tables

**Figure 1 biology-09-00327-f001:**
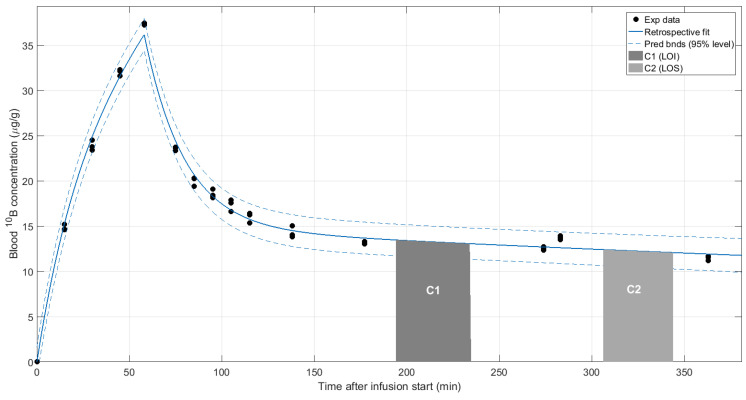
Blood ^10^B concentration–time profile after start of infusion. C1 and C2 correspond to the two portal irradiations.

**Figure 2 biology-09-00327-f002:**
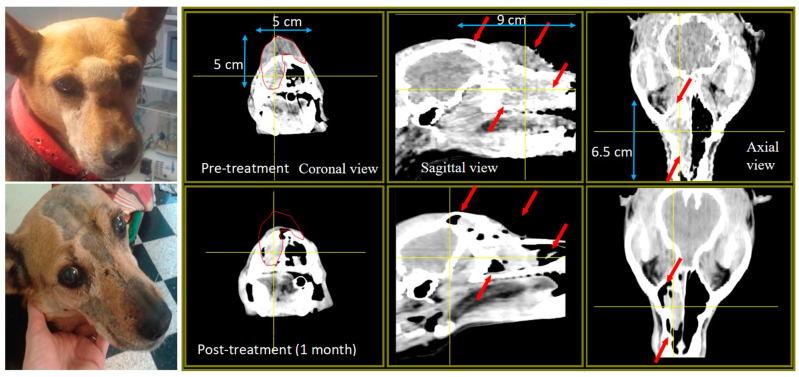
CT scan for Lucy pre-BNCT (top panel) and 1 month post BNCT (lower panel).

**Figure 3 biology-09-00327-f003:**
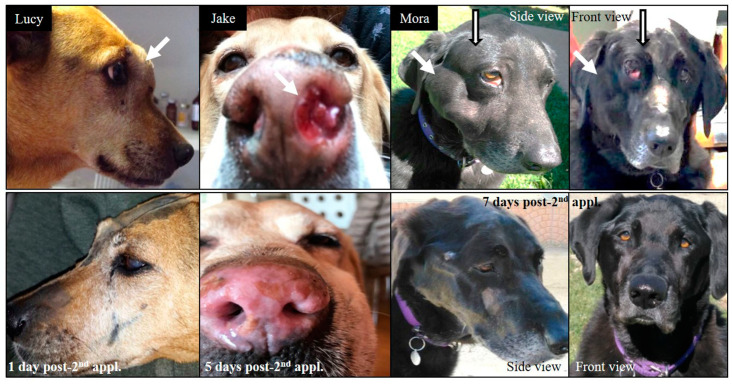
Patients pre-BNCT (top panel) and post BNCT (lower panel).

**Figure 4 biology-09-00327-f004:**
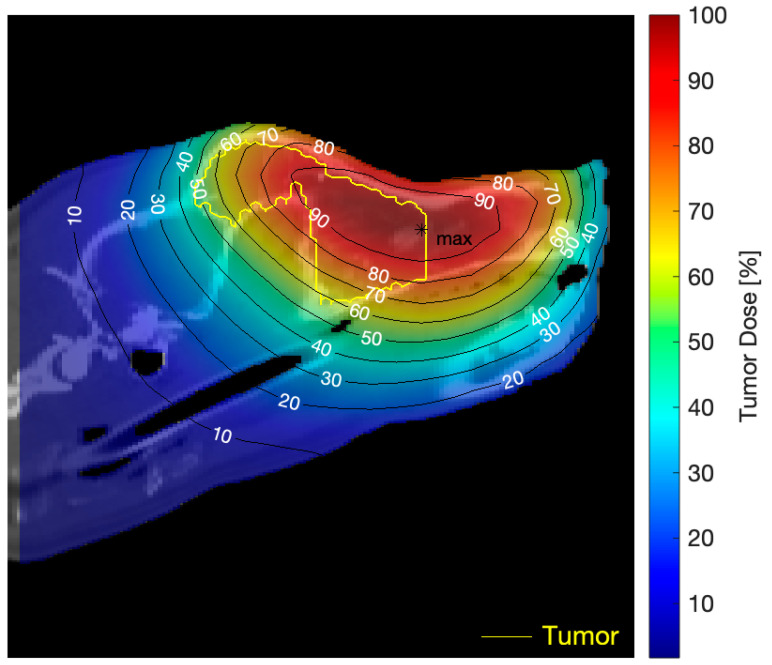
Example of tumor dose distribution (for Senshi, in this case) as a result of a three-portal irradiation per BNCT application employing the B2 beam. Tumor dose isocurves are overlaid onto a sagittal plane of the CT study.

**Table 1 biology-09-00327-t001:** Tumor doses for the 5 veterinary patients treated with the B2 clinical beam of the RA-6 reactor. BNCT1 and BNCT2 correspond to the first and second BNCT application (3–5 weeks apart), respectively.

		Tumor Total Doses, Mean (Min, Max)	
Dog	BNCT Application	Absorbed	RBE-Weighted
**Lucy**	BNCT1	5 (3, 6)	13 (8, 15)
	BNCT2	6 (4, 8)	17 (10, 22)
**Senshi**	BNCT1	6 (3, 7)	16 (7, 20)
	BNCT2	6 (3, 7)	16 (7, 20)
**Mora I**	BNCT1	11 (7, 14)	28 (17, 37)
	BNCT2	3 (2, 4)	8 (4, 10)
**Mora II**	BNCT1	7 (5, 9)	17 (12, 22)
	BNCT2	6 (5, 8)	17 (12, 22)
**Jake**	BNCT1	10 (7, 11)	26 (18, 31)
	BNCT2	10 (7, 11)	26 (18, 31)

**Table 2 biology-09-00327-t002:** Normal tissue doses for the 5 veterinary patients treated with the B2 clinical beam of the RA-6 reactor. BNCT1 and BNCT2 correspond to the first and second BNCT application (appl.) (3–5 weeks apart), respectively.

Normal Tissue Total Doses, Mean (Min, Max)						
	RBE-Weighted					Absorbed (Max)
Dog	BNCT Appl.	Left Eye Dmax = 6.5 Gy-eq	Right Eye Dmax = 6.5 Gy-eq	Brain Dmean = 7 Gy-eq	Skin Dmax = 22 Gy-eq	Mucosa Gy
**Lucy**	BNCT1	4.3 (3.6, 4.9)	4.3 (3.6, 5.0)	2.8 (1.1, 5.1)	3.0 (0.2, 7.7)	3.6
	BNCT2	4.3 (3.5, 5.0)	5.7 (5.4, 6.2)	3.4 (1.6, 5.6)	4.3 (0.2, 10.6)	5.0
**Senshi**	BNCT1	5.9 (5.2, 6.5)	5.5 (5.0, 6.0)	2.3 (0.9, 5.5)	3.5 (0.2, 9.8)	4.7
	BNCT2	6.1 (5.4, 6.7)	5.7 (5.2, 6.2)	2.3 (0.9, 5.7)	3.6 (0.2, 10.0)	4.9
**Mora I**	BNCT1	4.5 (3.2, 5.5)	9.4 (7.9, 10.4)	6.0 (3.8, 8.2)	6.8 (0.7, 17.6)	9.3
	BNCT2	1.5 (1.1, 1.8)	2.7 (2.3, 3.2)	1.9 (1.2, 2.7)	1.9 (0.2, 5.1)	2.6
**Mora II**	BNCT1	2.3 (1.8, 2.8)	4.7 (3.6, 5.6)	3.1 (1.8, 5.0)	4.6 (0.9, 7.4)	6.9
	BNCT2	1.9 (1.5, 2.3)	4.0 (3.1, 4.8)	2.5 (1.4, 4.2)	4.0 (0.4, 10.5)	6.7
**Jake**	BNCT1	4.3 (3.1, 5.7)	4.0 (3.0, 5.4)	1.5 (0.7, 4.0)	3.4 (0.4, 15.3)	7.7
	BNCT2	4.3 (3.1, 5.7)	4.0 (3.0, 5.4)	1.5 (0.7, 4.0)	3.4 (0.4, 15.3)	7.7

**Table 3 biology-09-00327-t003:** Patient details and outcome. SCC: Squamous Cell Carcinoma; BPA-BNCT: BNCT mediated by the boron compound BPA.

Patient	Histology/Pre-Treatment Tumor Volume (TV)	Complementary Treatment	BPA-BNCT	Response	Toxicity	Clinical Evaluation
**Mixed-breed** **Lucy** **12 years old**	SCC nasal cavity and paranasal sinuses with bone destruction Pre-treatment TV: 51 cm^3^ Re-treatment Pre re-treatment TV: 77 cm^3^	Chemo-therapy pre-BNCT	2 appl., 3 weeks apart; 2 portals per appl. Re-treatment with full dose BNCT, 10 months after the first BNCT treatment, due to local tumor regrowth	Partial response (PR) TV 1 month post 1st treatment BNCT: 25 cm^3^ Cause of death: euthanasia due to recurrence and decline Survival post 1st treatment: 13.5 months Survival post re-treatment: 3 months	Mild nasal keratosis Mild mucositis Mild somnolence Re-treatment: Mild somnolence and eye irritation	Positive tumor response and clinical benefit For 8 months, optimum clinical signs and no protruding tumor mass. Re-treatment due to tumor regrowth at 10 months post 1st treatment Initial positive response, recurrence and decline 3 months after re-treatment
**Alsatian** **Senshi** **9–10 years old**	Nasal chondrosarcoma TV: 105 cm^3^	Chemo-therapy pre-BNCT	2 appl., 3.5 weeks apart; 3 portals per appl.	PR TV 1 month post-BNCT: 57 cm^3^ Cause of death: euthanasia due to recurrence and decline Survival post BNCT: 10 months	Mild somnolence Mild–moderate nasal mucositis	Positive tumor response and clinical benefit
**Labrador Mora I** **9 years old**	Oral amelanotic melanoma TV: 342 cm^3^	5 surgeries Immuno-therapy pre-BNCT	2 appl., 5 weeks apart; 2 portals/appl.	PR TV 1 month post 1st appl.: 284 cm^3^ TV 2.5 months post 1st appl.: 149 cm^3^ Cause of death: Euthanasia due to lung metastasis diagnosed 2 months post BNCT Survival post 1st appl.: 2.5 months	Moderate mucositis	Positive tumor response and clinical benefit
**Labrador** **Mora II** **11 years old**	Oral amelanotic melanoma TV: 12 cm^3^	Surgery pre-BNCT Immuno-therapy 10 weeks post BNCT when liver and lung metastases were diagnosed	2 appl., 4.5 weeks apart; 2 portals per appl.	PR TV 1 month post 1st appl.: 2.9 cm^3^ TV 2 months post 1st appl.: 0.60 cm^3^ Cause of death: recurrence and lung metastasis Survival post BNCT: 1 year	Mild mucositis	Positive but short-term tumor response and clinical benefit
**Labrador Jake** **12 years old**	Nasal SCC (poorly differentiated) Myasthenia TV: 29 cm^3^	1 surgery pre-BNCT Palliative chemo-therapy 7 months post BNCT, after regrowth	2 appl., 4.5 weeks apart; 2 portals per appl.	PR as revealed by complete reduction in visible tumor volume and improvement in quality of life Cause of death: euthanasia due to recurrence and decline No CT scan 1 month post treatment Survival: 8.5 months post-BNCT	Moderate–severe mucositis that responded to medication	Positive tumor response and clinical benefit

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
