# Peer review of "Clinical Veterinary Boron Neutron Capture Therapy (BNCT) Studies in Dogs with Head and Neck Cancer: Bridging the Gap between Translational and Clinical Studies"

_biology, 2020, doi:10.3390/biology9100327_

Round 1
Reviewer 1 Report
General Comments
This article addresses the possibility and feasibility of application of BNCT for veterinary medicine. The detailed descriptions of post clinical course of five dogs treated with BNCT (Table 2) have informative contents for veterinarians who are not familiar with BNCT.
Specific comments.
Materials and methods
I fully understand the difficulties and complexity in reporting the doses delivered to the tumor or normal tissues in BNCT. Therefore, the authors should explain how the doses are estimated in more detail.
- P3 line 120
The authors should spell out the full term of RBE and CBE at the first time.
- P3 line 120
The authors should explain the definition of CBE for readers unfamiliar with BNCT.
- P3 line 121
The figure of 4.9 is adopted as the CBE factor for mucosal membrane in many BNCT research articles. I think that the authors explain why the figure of 2.5 was adopted as the CBE factor for mucosal membrane.
- P3 line 123-125.
The authors should explain why the upper limit dose for mucosa was physical dose (Gy) while the doses for other normal tissues were estimated in RBE-weighted dose (Gy-eq).
- P4 line 154
The BNCT radiation field consists of mixed radiation components. The authors should describe the component or gamma-ray.
Results
- The authors should show the dose parameters (Min, Mean, Max) in 5 dogs in the Results. In Table 1a and 1b, the doses delivered to tumor and normal tissues for Lucy was shown. Since there would be a correlation between the irradiated dose and outcomes of tumor response or adverse events, the authors summarize the dose in 5 dog patients.
- In connection with 1, the authors should show the dose distribution in the tumor with isodose lines overlaying in the CT images. The authors discussed that the underdose for deep-seated tumors leading to re-growth. It is important to show what part of the tumor was low-dose region by BNCT. Most of the readers do not know the rapid attenuation of the neutron beam in the body.
- The authors show the profile of boron concentration in the blood during irradiation. The boron concentration in the blood rapidly deceases after the end of intra-venous administration of BPA since in this treatment protocol, BPA administration was finished before irradiation.
Discussion
- The authors should discuss the reason why complete remission was experienced in this study although, in general, the tumors in dogs and cats are smaller than those in human.
2, P9 line 267
The authors should spell out the full term of LAT at the first time.
Author Response
Specific comments.
Materials and methods I fully understand the difficulties and complexity in reporting the doses delivered to the tumor or normal tissues in BNCT. Therefore, the authors should explain how the doses are estimated in more detail.
1.P3 line 120, The authors should spell out the full term of RBE and CBE at the first time.
The terms were written out in full.
2. P3 line 120, The authors should explain the definition of CBE for readers unfamiliar with BNCT.
A brief explanation of the concept of CBE has been included.
3. P3 line 121, The figure of 4.9 is adopted as the CBE factor for mucosal membrane in many BNCT research articles. I think that the authors explain why the figure of 2.5 was adopted as the CBE factor for mucosal membrane.
We have followed the Finnish protocol used for the treatment of HN cancer patients. Within this protocol, a mucosa-to-blood 10B concentration ratio of 2.0 and a CBE factor for mucosal membrane of 2.5 are assumed. Some BNCT groups decide to use a CBE factor of 4.9 but a mucosa-to-blood 10B concentration ratio of 1.0. In this regard, the absorbed boron dose component in units of Gy/ppm is finally weighted by almost the same factor, i.e. 2.0 x 2.5 = 5 in our case, and 4.9 for the other groups.
This has been explained briefly in the text.
4. P3 line 123-125, The authors should explain why the upper limit dose for mucosa was physical dose (Gy) while the doses for other normal tissues were estimated in RBE-weighted dose (Gy-eq).
We originally adopted the different upper limits included in the Finnish protocol to determine the irradiation times in the clinical-veterinary treatments. In this protocol, the maximum dose in mucosa is limited to 6 Gy (absorbed dose) and the remaining upper limits are established in RBE-weighted doses.
However, based on our NTCP model for mucositis G3 or higher that depends on the four absorbed doses delivered in BNCT rather than on RBE-weighted doses, we determined that 6 Gy of total absorbed dose administered with our B2 clinical beam gives a very low probability of radiotoxic effects (NTCP < 0.04). Since FIR1 clinical beam has different relative components compared to B2 clinical beam, the same physical dose of 6 Gy delivered with FIR1 gives a higher NTCP value of about 0.5. Therefore, we decided to start by limiting the absorbed dose to the mucosa to 6 Gy and to then scale this limit based on our NTCP model and a case by case clinical scenario.
This issue has been addressed in the text and the corresponding reference has been added (González et al. 2017, reference 27).
5. P4 line 154, The BNCT radiation field consists of mixed radiation components. The authors should describe the component or gamma-ray.
The in-air thermal and epithermal neutron fluxes estimated by computational analysis at the patient position and averaged over the circular aperture of 15 cm are, respectively, 2.47 x 108 n cm-2 s-1 and 2.16 x 108 n cm-2 s-1 with a statistical uncertainty of less than 1%. In addition, the specific gamma dose (i.e., ) is 1.74 x 10-12 Gy cm2 n-1.
This has been clarified in the text.
Results
1. The authors should show the dose parameters (Min, Mean, Max) in 5 dogs in the Results. In Table 1a and 1b, the doses delivered to tumor and normal tissues for Lucy was shown. Since there would be a correlation between the irradiated dose and outcomes of tumor response or adverse events, the authors summarize the dose in 5 dog patients.
Table 1a now corresponds to Tumor doses for the 5 veterinary patients treated with the B2 clinical beam of the RA-6 reactor.
Table 1b now corresponds to the Normal tissue doses for the 5 veterinary patients treated with the B2 clinical beam of the RA-6 reactor.
2. In connection with 1, the authors should show the dose distribution in the tumor with isodose lines overlaying in the CT images. The authors discussed that the underdose for deep-seated tumors leading to re-growth. It is important to show what part of the tumor was low-dose region by BNCT. Most of the readers do not know the rapid attenuation of the neutron beam in the body.
We have now included Figure 3 that shows, as an example, the tumor dose distribution for Senshi as a result of a three portal irradiation per BNCT application. Tumor dose isocurves are overlaid onto a sagittal plane of the CT study.
The fact that the mean tumor dose is close to the maximum value (Table 1a) confirms that a significant portion of the tumor volume received high doses. However, since the tumor is deep-seated and the thermal neutron flux of the B2 mixed beam peaks at about 1 cm depth and then decreases, the most distal parts of the lesion received only moderate doses that would not be enough to induce complete tumor remission.
3. The authors show the profile of boron concentration in the blood during irradiation. The boron concentration in the blood rapidly deceases after the end of intra-venous administration of BPA since in this treatment protocol, BPA administration was finished before irradiation.
We have added Figure 1 (Blood 10B concentration – time profile after start of infusion. C1 and C2 correspond to the two portal irradiations) that depicts an example of our general treatment setting. As can be observed in the figure, the blood boron concentration profile shows a bi-exponential decay characterized by a fast and a slow component with different half-life values. Portal irradiations are always performed when the slow component dominates, thus reducing boron concentration changes while maintaining acceptable absolute values.
The remaining figures have been renumbered.
Discussion
1. The authors should discuss the reason why complete remission was experienced in this study although, in general, the tumors in dogs and cats are smaller than those in human.
Although the number of dogs treated is insufficient to draw conclusions in this sense, irradiation with the B2 beam, designed to treat superficial tumors, might lead to underdosing of certain tumor areas and incomplete tumor remission.
This has been clarified in the text.
2. P9 line 267, The authors should spell out the full term of LAT at the first time.
The term has been written out in full.
Reviewer 2 Report
In this manuscript, the authors provide a great contribution to the field of BNCT by assessing the response to this type of cancer therapy of five dogs with spontaneous head and neck tumors. The article is well written and provides useful and valuable information for the improvement and application of BNCT not only in veterinary but also in human medicine, as the animal model of the study mimics the features of human cancers. Moreover, it is also relevant for nuclear physicists who can progress in the development of better neutron beams for clinical applications. The possibility of using existing nuclear reactors for treatment of deep-seated tumors by BNCT or designing new ones for this application is of great interest for the advance of cancer therapy. I therefore recommend the publication of this manuscript in Biology.
I have some minor comments:
Page 3, line 97: There are many types of lymphomas and it is not always a systemic disease. It often appears in localized areas, such as lymph nodes. Hence, it should not be cited in general as a systemic disease.
Page 3, line 107: “Life expectancy with acceptable quality of life (QoL) of >1-2 months”, I would say “… of ≤ 1-2 months”
It should be interesting to provide a bit more explanation about the clinical impact of BNCT and the clinical development of patients. Why immunotherapy was administered 10 weeks post-BNCT to Labrador Mora II and how it should influence the evolution of this patient? When lung metastasis were detected in the patients?
There is no data of CT post-treatmet for Jake. How do you prove the positive response in this case?
Author Response
Page 3, line 97: There are many types of lymphomas and it is not always a systemic disease. It often appears in localized areas, such as lymph nodes. Hence, it should not be cited in general as a systemic disease.
The referee is quite right. Thank you for making this point. The reference to lymphoma as a systemic disease has been deleted.
Page 3, line 107: “Life expectancy with acceptable quality of life (QoL) of >1-2 months”, I would say “… of ≤ 1-2 months”
Our aim was to include dog patients that were well enough to allow for minimum follow-up and were well enough to withstand treatment that involved approximately 2 hours of general anesthesia. That is why our inclusion criteria included “Life expectancy with acceptable quality of life (QoL) of >1-2 months” (more than 1-2 months).
In addition, our aim was to include patients that did not have any other therapeutic option, regardless of the expected time of survival (provided they fulfilled the criterion “Life expectancy with acceptable quality of life (QoL) of >1-2 months”). Based on our knowledge, we did not expect BNCT to reduce survival time. Our working hypothesis was that BNCT would extend survival time. The way it turned out, the animals that were recruited were well enough to withstand prolonged anesthesia but the veterinarians anticipated that they would have to be euthanized 1-2 months later due to anticipated decline (if they were not treated with BNCT). However, this time frame (less than 2 months anticipated survival) was not an inclusion criterion.
This issue has been briefly clarified in the text to avoid confusion.
It should be interesting to provide a bit more explanation about the clinical impact of BNCT and the clinical development of patients. Why immunotherapy was administered 10 weeks post-BNCT to Labrador Mora II and how it should influence the evolution of this patient? When lung metastasis were detected in the patients?
Immunotherapy was administered to Mora II ten weeks post BNCT because at that time liver and lung metastases were diagnosed by a second follow-up TAC. (I hope you understand that the possibility of performing TAC was not the same for all patients). Immunotherapy conceivably contributed to stabilize this patient in terms of clinical signs.
Lung metastases were diagnosed in Mora I two months post BNCT. Shortly afterwards she developed breathing difficulties and was euthanized 2 weeks later. This was a very disappointing outcome because local response was excellent, allowing her to eat normally and play.
This information was summarized in Table 2.
There is no data of CT post-treatmet for Jake. How do you prove the positive response in this case?
As indicated in Table 2, in this case it was not possible to perform a post-treatment CT as with the other dogs. Response was evaluated by visual inspection and palpation of tumor that was amenable to macroscopic assessment. We observed an ostensible reduction in visible tumor volume (Figure 2) that could not be quantified and a very marked improvement in clinical signs (mainly breathing). Based on this evaluation, response was classified as “partial response” even without a TC. This has been briefly clarified in Table 2.
Reviewer 3 Report
This paper presents the process and outcome of BNCT treatment conducted on five dogs with terminal H&N cancer (QoL of 1 to 2 months). Treatment have been delivered in 2 to 3 fractions, and in all cases the authors have reported a positive response (reduction of tumour volume by ~50%) and an increased QoL.
This is an interesting work, and I would like to congratulate the authors in conducting the experiments, especially given the logistical difficulties (including the remote monitoring of the animals throughout the 2 hour treatment delivery).
I have a number of general and specific comments:
- The paper follows a logical template, however it is missing a Conclusion section. I recommend that the authors add this after their Discussion Section.
- The authors have claimed that these animals had no other treatment options. Stereotactic radiation therapy (SRT) is a curative form of treatment which delivers a conformal gamma dose using a number of modalities (cyberknife, IMRT etc) [https://www.ncbi.nlm.nih.gov/pmc/articles/PMC4266068/]. In more recent times, the use of high intensity spatially fractionated radiation (such as microbeam radiation therapy) has been explored for the treatment of canine sarcomas [https://journals.sagepub.com/doi/full/10.1177/1533034617690980].
Whilst I agree that the outcome of this study is promising, it lacks a control group to evaluate its efficacy against SRT or MRT. In fact, this is one of the main criticisms of the radiation oncology community with regards to BNCT - that is, the scarcity of controlled preclinical and clinical data.
The authors can for instance add a section to their paper, use a TPS for planning the delivery of an equivalent dose using SRT, and through employing statistical models, predict the resulting QoL and NTCP. Alternatively, this work can be rephrased as an exploratory study, in preparation for future controlled studies. - Discussions, Line 258-259: The authors attribute the eventual tumour regrowth to an underdosing of the distal part of the deeply seated tumours. They have remarked that this can be resolved by "modifications to obtain more penetrating beam". Are the authors proposing the modification of beam flux, or the beam spectrum? Either of the two will result in an increased dose in the normal tissue proximal to the tumour (increased skin and blood dose). I suggest that the authors provide more details around the proposed modifications.
- Discussions, Line 267 - 269: The combination of specific and non specific boron delivery - while increasing the tumour boron concentration - can potentially reduce the T/N and T/B boron ratios. Please comment on the potential increased NTCP.
- Line 131, Table 1a: How was the tumour ppm measured (tissue sample during the procedure?)
- Editorial comment: Lines 45, 48, 49 (and other instances): Please remove "e.g." from citing your references.
Author Response
1. The paper follows a logical template, however it is missing a Conclusion section. I recommend that the authors add this after their Discussion Section.
A Conclusion section was added as suggested.
2. The authors have claimed that these animals had no other treatment options. Stereotactic radiation therapy (SRT) is a curative form of treatment which delivers a conformal gamma dose using a number of modalities (cyberknife, IMRT etc) [https://www.ncbi.nlm.nih.gov/pmc/articles/PMC4266068/]. In more recent times, the use of high intensity spatially fractionated radiation (such as microbeam radiation therapy) has been explored for the treatment of canine sarcomas [https://journals.sagepub.com/doi/full/10.1177/1533034617690980].
Whilst I agree that the outcome of this study is promising, it lacks a control group to evaluate its efficacy against SRT or MRT. In fact, this is one of the main criticisms of the radiation oncology community with regards to BNCT - that is, the scarcity of controlled preclinical and clinical data.
The authors can for instance add a section to their paper, use a TPS for planning the delivery of an equivalent dose using SRT, and through employing statistical models, predict the resulting QoL and NTCP. Alternatively, this work can be rephrased as an exploratory study, in preparation for future controlled studies.
We fully agree with the point made by the referee. Unfortunately, SRT and MRT are not available for veterinary patients in our country. In this sense, this would be an exploratory study providing contributory data for other centers worldwide which might be able to perform controlled studies. I am aware of groups in Japan interested in BNCT for veterinary medicine that might want to perform this sort of study in the future. The exploratory nature of our study has now been stated in the text.
3. Discussions, Line 258-259: The authors attribute the eventual tumour regrowth to an underdosing of the distal part of the deeply seated tumours. They have remarked that this can be resolved by "modifications to obtain more penetrating beam". Are the authors proposing the modification of beam flux, or the beam spectrum? Either of the two will result in an increased dose in the normal tissue proximal to the tumour (increased skin and blood dose). I suggest that the authors provide more details around the proposed modifications.
Different beam shaping assembly models and configurations, including delimiters and shields have been evaluated, all of them aimed at modifying the neutron beam spectrum of the BNCT facility of the RA-6 reactor to obtain an epithermal beam, with a maximum yield of thermal neutrons at greater depths. The computational models also aim at providing a tunable beam by means of moderating interchangeable materials to lower the neutron energies and maintain the original spectrum, originally optimized to treat shallow lesions.
Epithermal neutron spectra used in BNCT clinical trials inherently provide lower doses in the first few millimeters, since the neutron energies at the entrance in tissue are those where a natural minimum in the total neutron interaction cross-section (and in consequence in the total KERMA factor) exists. Actually, an epithermal neutron beam delivers lower doses at the entrance compared to a thermal or hyperthermal beam.
Additional information on the proposed beam modifications has been included in the manuscript as requested.
4. Discussions, Line 267 - 269: The combination of specific and non specific boron delivery - while increasing the tumour boron concentration - can potentially reduce the T/N and T/B boron ratios. Please comment on the potential increased NTCP.
The reviewer is quite right. In the case of GB-10, tumors do not incorporate boron preferentially in the hamster cheek pouch oral cancer model. However, our studies in this model with BNCT mediated by GB-10 and by GB-10+BPA showed a new paradigm in BNCT (Trivillin et al. Rad Res 2006). Despite the fact that GB-10 does not target hamster cheek pouch tumors selectively, GB-10-BNCT induced a 70% overall tumor response with no damage to normal tissue and only mild mucositis in precancerous tissue (the dose-limiting tissue in this model). (GB-10+BPA)-BNCT induced a 93% overall tumor response with no normal tissue radiotoxicity and only mild mucositis in precancerous tissue. Although BPA-BNCT induced 91% tumor response, mucositis in precancerous tissue was severe. Light microscope analysis showed that GB-10-BNCT selectively damages the more radiosensitive aberrant tumor blood vessels, while sparing precancerous and normal tissue blood vessels.
In the case of GB-10-BNCT, selective tumor lethality would result from selective tumor blood vessel damage rather than from selective tumor uptake of the boron compound.
This has been briefly explained in the text.
5. Line 131, Table 1a: How was the tumour ppm measured (tissue sample during the procedure?)
Blood boron concentration during and after infusion is measured by ICP-OES (Inductively-Coupled Plasma Optical Emission Spectroscopy) and an open 2-compartment model is used to fit the data and predict boron concentration in blood during the irradiation. Tumor boron concentration, however, is not measured (no tissue samples are taken during treatment). To predict the boron dose to tumor, tumor-to-blood (T/B) ratios are adopted and used. In our case, a T/B value of 3.0 is assumed as a conservative criterion, slightly less than the value used in the Finnish BNCT clinical protocol (T/B = 3.5) and in keeping with translational studies performed by our group in the hamster cheek pouch oral cancer model .
6. Editorial comment: Lines 45, 48, 49 (and other instances): Please remove "e.g." from citing your references.
As requested, we removed “e.g.” in all the cases where we cited our own references.
Reviewer 4 Report
I enjoyed reading this manuscript, and I was very pleased to rnow that BNCT was again used in veterinary medicine for the treatment of dogs, since I was well familar with the participants in a similar project, interrupted by the premature death.
The manuscript contains important new information, is well written and undoubtedly deserves to be published in the special issue of Biology devoted to BNCT.
As a small note, I would like to advise the authors to add reference to the article S. L. Kraft, P. R. Gavin, C. W. Leathers, C. E. DeHaan, W. F. Bauer, D. L. Miller, R. V. Dorn III, M. L. Griebenow, Biodistribution of Boron in Dogs with Spontaneous Intracranial Tumors following
Borocaptate Sodium Administration, Cancer Res., 1994, 54, 1259-1263.
Author Response
As a small note, I would like to advise the authors to add reference to the article S. L. Kraft, P. R. Gavin, C. W. Leathers, C. E. DeHaan, W. F. Bauer, D. L. Miller, R. V. Dorn III, M. L. Griebenow, Biodistribution of Boron in Dogs with Spontaneous Intracranial Tumors following Borocaptate Sodium Administration, Cancer Res., 1994, 54, 1259-1263.
I would like to thank our reviewer for his/her very kind words. We have added the very contributory reference he/she calls our attention to and apologize for not including it before.